

# Monitoring ephemeral, intermittent and perennial streamflow: A data set from 182 sites in the Attert catchment, Luxembourg

Nils H. Kaplan[1], Ernestine Sohrt[2], Theresa Blume[2], Markus Weiler[1]

[1]Hydrology, Faculty of Environment and Natural Resources, University of Freiburg, 79098 Freiburg, Germany
5  [2]Hydrology, Helmholtz Centre Potsdam, GFZ German Research Centre for Geosciences, 14473 Potsdam, Germany

*Correspondence to*: Nils H. Kaplan (nils.kaplan@hydrology.uni-freiburg.de)

**Abstract.** The temporal and spatial dynamics of streamflow presence and absence is considered vital information to many hydrological and ecological studies. Measuring the duration of active streamflow and dry periods in the channel allows us to classify the degree of intermittency of streams. We used different sensing techniques including time-lapse imagery, electric conductivity and stage measurements to generate a combined dataset of presence and absence of streamflow within various nested sub-catchments in the Attert Catchment, Luxembourg. The first sites of observation were established in 2013 and successively extended to a total number of 182 in 2016 as part of the project "Catchments As Organized Systems" (CAOS). Temporal resolution ranged from 5 to 15 minutes intervals. Each single dataset was carefully processed and quality controlled before the time interval was homogenised to 30 minutes. The dataset provides valuable information of the dynamics of a meso-scale stream network in space and time. This can be used to test and evaluate hydrologic models, but also for the assessment of the intermittent stream ecosystem in the Attert basin. The dataset presented in this paper is available at the online repository of the German Research Center for Geosciences (GFZ, http://doi.org/10.5880/FIDGEO.2019.010).

## 1 Introduction

20  Based on the permanence of flow, streams can be classified into ephemeral, intermittent or perennial using hydrological (e.g. Hedman & Osterkamp, 1982; Uys & O'Keeffee, 1997; Svec et al. 2005, Wohl 2017) or ecological (e.g. Hansen 2001; Leigh et al., 2015; Stromberg & Merritt, 2015) indicators. Statistical indicators solely based on the duration of stream flow allow flexible categorisation of streams according to the predominant annual average climatic conditions. For the Western United States Hedman & Osterkamp (1982) define streams with measureable annual surface flow for more than 80 % of the time as 25  perennial, whereas streams with surface flow occurring between 80 % and 10 % of the year are categorized as intermittent and as ephemeral when the duration of streamflow per year falls below 10 %. Slightly different classes can be found for North American prairie streams at Matthews (1988) who classified streams with flow not exceeding 20 % of the year as ephemeral, intermittent with flow between 20 % to 80 % of the year and perennial with flow for more than 80 % of the year. Streamflow classification can also be based on hydrological processes involved to maintain flow and the type of sources



which provide water to the stream over the year (e.g. Sophocleous, 2002; Nadeau & Rains, 2007). Under normal conditions perennial streams flow throughout the year and receive substantial amounts of water from various sources including snowmelt and effluent groundwater and sometimes snowmelt (Sophocleous, 2002; Nadeau & Rains, 2007). Intermittent streams cease to flow during dry periods when the state changes from gaining groundwater (effluent) to losing groundwater

(influent) while the source of streamflow can originate from various sources (Sophocleous, 2002; Nadeau & Rains, 2007). During dry periods of the year intermittent streams can dry out completely or partially, i.e. along certain stream segments which are then disconnected from the longitudinal flow (Wohl, 2017). In ephemeral streams the groundwater level is always lower than the channel (influent) and therefore flow only occurs in direct response to precipitation (Sophocleous, 2002). While public agencies rarely monitor discharge in intermittent and ephemeral streams (Uys & O'Keeffe, 1997; Svec et al.

2002), the dynamics of the total stream network and therefore reliable information on the temporal dynamics of the spatial extent is nevertheless of great interest to many hydrological studies (e.g. Godsey & Kirchner, 2014; Shaw, 2016; Stoll & Weiler, 2010; Jensen et al., 2017). Several methods can be used to obtain information about the presence of flow in stream networks ranging from point measurements at conventional gauging systems or flow monitoring sensors over line feature information through visual assessment during field surveys to spatial information from remote sensing products. Stoll and

Weiler (2010) propose to use remote sensing products as ground truth to verify simulated spatial dynamics of stream networks in ungauged basins. Remote sensing products usually suffer from low temporal resolution on the one hand (as a result of the return interval of the satellites as well as data gaps caused by cloud cover) and from total lack of information for tree-shaded stream reaches on the other hand.

Field surveys of wet vs dry and subsequent mapping are commonly used to monitor the spatial extent of the stream network

(e.g. Godsey and Kirchner, 2014; Shaw, 2016; Goodrich et al., 2018). This is labour intensive and hence the mapping is usually carried out either only seasonally (Godsey and Kirchner, 2014), in uneven intervals during the dry season and directly after strong rainfall events (Day, 1978; Davids et al., 2017) or dependent on the discharge of a reference gauge (Jensen et al., 2017). Recently citizen science approaches are being developed for wet/dry mapping in order to minimize labour intensive fieldwork (Turner and Richter, 2011; Davids et al., 2017).

Various measurement methods of streamflow at the point scale were established for continuous, automated recording of stage height at weirs or flumes at perennial streams (e.g. Sauer and Turnipseed, 2010). While pressure transducers have been used in studies of temporary stream monitoring (Gupta R., 2001; Svec et al. 2005), this monitoring method has some drawbacks: the determination of accurate rating curves and thus correct measurements of zero flow in temporal streams can be challenging due to the unpredictability of the active flow channels within the streambed and often significant erosion or

deposition (Shanafield and Cook, 2014). Pressure transducers are also considered to be expensive in cases were the aim of monitoring is not discharge but streamflow duration (Bhamjee and Lindsay, 2011). Monitoring of streambed temperature is a robust and inexpensive method to differentiate between presence and absence of streamflow and can help to overcome the financial drawbacks of conventional gauging systems (e.g. Constantz et al. 2001, Selker et al. 2006, Buttle et al 2013). Constantz et al. (2001) collected streambed temperature data in longitudinal stream transects in order to monitor the spatial



and temporal patterns of streamflow in ephemeral channels. However the sensors needed to be constantly calibrated to air temperature in situations where the streambed and thus the temperature sensor can fall dry (Constantz et al. 2001). Buttle et al. (2013) identify the point-scale of a single temperature sensor as a limitation of the method. This can be overcome by fibre optic temperature sensing for continuous monitoring of vertical and longitudinal profiles of stream temperatures (Selker et al., 2006).

Point scale electric resistance (ER) sensors can provide more accurate estimation of the occurrence of streamflow duration requiring less interpretation compared to temperature measurements (Blasch et al. 2002). ER sensors have been used in many studies to deduce the temporal and spatial dynamics of the occurrence of streamflow in different environments including dry channels and wetlands (e.g. Adams et al., 2006; Goulbra et al., 2009; Bhamjee and Lindsay, 2011; Chapin et al., 2014) and the connectivity of rivers (Jaeger and Olden, 2012). In respect to timing of streamflow the achieved accuracy of the ER sensors was comparable to stream gauge measurements (Blasch et al. 2002). Bhamjee et al. (2016) propose a paired sensor approach combining an ER-sensor and a flow-detection sensor, which is based on a flap that opens under water pressure, in order to overcome the limitation in distinguishing between pooling water and flowing water as well as the dry state of the channel.

In recent years the use of image based methods to identify flow velocity (e.g. Bradley et al., 2002; Muste et al., 2008, Tsubaki et al., 2015), stage (e.g. Shin et al., 2007; Royem et al., 2012; Gilmore et al., 2013, Schoener 2017) or discharge (Luethi et al., 2014) found its way into hydrological applications. Streamflow measurements using particle image velocimetry (PIV) requires video imagery and has been used in combination with artificial and natural tracers (Bradley et al., 2002; Creutin et al., 2003; Muste et al., 2008). Implementation of PIV in applications for mobile devices enables the use as mobile measurement device for streamflow (Tsubaki et al. 2015). Luethi et al. (2014) used PIV in a smartphone application in combination with derived water level and measured channel geometry to calculate discharge.

Besides information on streamflow velocity, image based approaches can also be used to measure water level. Non-contact liquid level measurements using computer vision have been proposed by Chakravarthy et al. (2002) for fuel tank application. Shin et al. (2007) developed a stage measurement method based on images of a gauging staff in a stream. Their image processing is based on the assumption that camera and staff gauge do not move and therefor the staff gauge has an identical position in all images which can be defined by a region of interest (ROI). They were able to detect water level after image processing with a mean difference of 2 % between automated measured and visually measured pixels that indicate water level height. Royem et al. (2012) showed the value of wildlife cameras with time lapse function as potentially reliable tool to enable stream monitoring in citizen science. They combined the camera with a bright yellow steel ruler as gauging staff. Evaluation of measured heights from the camera system with stage heights from the United States Geological Survey (USGS) showed a good agreement with a relative difference of 16%. Gilmore et al. (2013) identified image resolution, lighting effects, perspective, lens distortion and the water meniscus as sources of error. They found image resolution and the meniscus contributing most to errors in detected water level, while the influence of lens distortion largely depends on the consideration of the distortion in the software. Image based stage recording can provide backup information to existing



gauging stations (SEBA- Hydrometrie, 2017). It can also be a practical and cheaper alternative to conventional stream gauging for the temporary monitoring of ephemeral and intermittent streams where costs of installing a conventional gauging system are not reasonable (Peters et al., 2013; Schoener, 2017). Despite these advances in measurement methods, designing monitoring networks for the dynamics of stream network extend remains challenging. According to Bhamjee et al.

(2016) determining the adequate spacing of sites along a stream can be difficult as channel and flow characteristics can change over short distance. This can lead to misclassification of streamflow intermittency due to point scale information and thus incomplete monitoring of linear features.

Setting up an affordable and dense monitoring system for a meso-scale catchment like the Attert requires the use of different

sensors for streams with different intermittency characteristics. Thus, we present in this paper streamflow data obtained by a variety of measurement sources including time-lapse imagery, EC-measurements and conventional water level gauging. Time-lapse imagery was primarily dedicated to measure streamflow in ephemeral to intermittent streams. EC-sensors on the other hand used to investigate streamflow duration within the intermittent to perennial reaches of the stream network. Compared to conventional stream gauging the information on water levels is lost but costs of installation and maintenance

are reduced. The sensor network was completed by conventionally gauged weirs which allow for evaluation of the data obtained by the EC-sensors and time-lapse cameras and to relate the observed pattern of streamflow occurrence to the integrated discharge signal. In total streamflow presence or absence was monitored at 182 locations, allowing us to observe the contraction and expansion of the stream network in the Attert basin over time.

**2 Site description**

The Attert basin is located in Luxembourg and Belgium and covers an area of 247 km² at the outlet at Useldange (Martinez-Carreras et al., 2012). With slate, marls and sandstone three prevalent geologies can be found in the catchment (Figure 1). In the north-western part of the catchment (24 % of the area) Devonian slate forms the Luxembourg Ardennes while the central part of the catchment is dominated by sandy Keuper-marls and the southern catchment boundary is defined by the Jurassic

Luxembourg Sandstone formation (Martinez-Carreras et al., 2012). Altitudes range from 245 m a.s.l. in Useldange to 549 m a.s.l. at the ridges of the Luxembourg Ardennes. Moderately sloping lowlands dominate the topography in the Keuper marls with steeper slopes at the hilly Luxembourg Sandstone formation (Martinez-Carreras et al., 2012). Land use in the areas of marls comprises a mixture of forest (29 %), grassland (26 %) and agriculture (41 %) with low proportions of urban areas (4 %) while the Sandstone areas are mainly forested (55 %) with lower proportions of grassland and agriculture (39 %). The

topography of the slate-rich Ardennes is characterized by plateaus with altitudes between 450 and 550 m a.s.l. and steep valleys incised down to 300 m a.s.l. The plateaus in the Ardennes are mainly used for agriculture (42 %) and urban areas (4 %) whereas slopes and valleys are dominated by forest (48 %) and pasture (6 %). The climate in the area is influenced by the



Atlantic Ocean and provides comparable conditions throughout the catchment. There is a slight spatial trend in mean annual precipitation with annual precipitation averages decreasing from about 1000 mm in the higher altitudes in the north-western part of the catchment to 800 mm in the south-eastern part (Pfister et al., 2017). Mean annual precipitation across the catchment was about 850 mm for the years 1971-2000 (Pfister et al., 2005) and showed on average low variability in

monthly precipitation sums ranging from 70 mm to 100 mm/month between driest and wettest month of the year (Wrede et al. 2014). Mean monthly temperatures show strong seasonal fluctuations and reach a maximum of 17°C in July and a minimum of 0°C in January (Pfister et. al., 2005). Consequently, monthly potential evapotranspiration follows the seasonal temperature changes with averages of around 13 mm/month in December up to around 80 mm/month in July (Wrede et al., 2014) adding up to yearly average sums of potential evapotranspiration of around 620 mm/y across the catchment (Pfister et

al., 2017). Therefore surface hydrology and the runoff regime is generally characterized by high, mostly rainfall driven flows during winter season and low flows during summer season as a result of higher evapotranspiration (Wrede et al., 2014). According to Pfister et al. (2017) bedrock geology has a strong influence on catchment storage, mixing and release of water in the Attert catchment which results in stronger differences between seasonal flow regimes in areas of impermeable bedrock (slate and marls) compared to regions with permeable sandstone bedrock or diverse geology. Even the topographic map

reveals the dependence of hydrological behaviour on bedrock permeability (Figure 1), with a lower drainage density in the sandstone regions (Le Gouvernment du Grand-Duché de Luxembourg, 2009). Surface flow in the area is impacted by different anthropogenic influences. Tile drains, installation of dams and ditches as well as other river regulation measures in the agricultural areas in the central marl dominated regions of the catchment lower the groundwater table and increase runoff velocity (Schaich et al. 2011). Thus, anthropogenic impacts can alter the behaviour of ephemeral and intermittent streams,

resulting in shorter periods of streamflow. Nevertheless, intermittent streams can also become potentially perennial if upstream water treatment plants provide constant flow of treated wastewater even in dry conditions which can be the case at plants located on the plateaus of the Ardennes (Le Gouvernment du Grand-Duché de Luxembourg, 2018).



**Figure 1: Geology and stream network of the Attert basin and streamflow monitoring sites. Monitoring sites are categorized into time-lapse Camera (C), Electric Conductivity measurements (EC) and Conventional discharge Gauging systems (CG). Detailed maps show the more densely equipped areas in each predominant geology: slate (blue box), marls (red box) and sandstone (green box). The geological map from 1947 was provided by the Geological Service of Luxembourg.**

## 3 Streamflow observations

### 3.1 Streamflow observation with time lapse photography

The time lapse photography system for streamflow observations is based on the consumer wild life camera Dörr Snapshot Mini 5.0 which was mounted on trees close to the stream and took images at 15 minutes intervals. The camera was mounted with a lashing strap to the tree, aligned with wooden wedges and secured with a cable lock. Access to the control panel of the camera and the 12GB SD-card was protected by a padlock. The camera was powered by a 12V external battery pack protected against wildlife and precipitation. A gauging plate was placed in the observed channels at the lowest point of the cross-section. The gauging plate with its layout of blue triangular markers on red coloured background was designed to





enable automatic water level extraction using image analysis (Figure 3). Energy consumption and storage capacity of the SD-card required a maintenance interval of two months.

We equipped 71 sites with time-lapse imagery streamflow observation mainly at smaller streams and channels with ephemeral to intermittent streamflow. Sites were selected and set up successively from August 2015 till July 2016. Data
collection ended in July 2017. The sites were selected to cover the diverse geology in the Attert basin in order to capture the temporal and spatial dynamics across the catchment. Site selection was with few exceptions constrained to be in public forestdue to the availability of permits. Forested sites also feature most of the intermittent streams in the Attert catchment. 40 sites were chosen at streams which are indicated as "intermittent" on the topographic map at the scale of 1:20.000 (Le Gouvernment du Grand-Duché de Luxembourg, 2009). Additional 21 sites at streams or channels which were not included
in the topographic map were identified by scouting close to existing sensor installations of the CAOS project (Zehe et al. 2014). Furthermore seven sites were selected from terrain analysis of a digital elevation model with a 5 m resolution at locations with high concave curvature and steep slopes. 3 sites at forest tracks in the Luxembourg sandstone in the southern part of the catchment were identified by washed out gully structures as part of the temporal stream network and therefore included in the observational network.

A plugin for the image processing software ImageJ (ImageJ Developers 2016) was further developed for pre-processing of images. Images with too much noise were automatically removed based on an empirically defined threshold of a maximum of 6 % pixels with the same colour value. Pre-processing included automated contrast enhancement allowing up to 5 % saturation, user defined cropping to region of interest and 3 sigma Gaussian filter to reduce scatter. Pre-processed images were loaded as virtual stacks into ImageJ and visually classified into images showing presence or absence of flow:
completely dry conditions lead to a classification of non-streamflow conditions and visible water in images were classified as streamflow conditions (Figure 2).

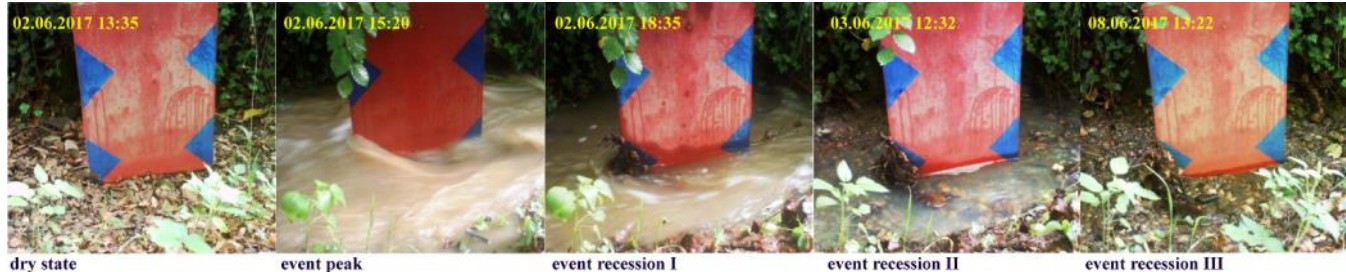

**Figure 2: Example of streamflow event classification (site ID: C34) with change from dry to wet state and back. The event starts between 13:35 and 15:20 on the 02.06.2017. Images taken between the time steps of dry conditions and event peak have been**
**automatically identified by the software as being too noisy. Flow decreases during the event recession and ends on the 10.06.2017.**



## 3.2 Streamflow observation with electric conductivity sensors

Presence or absence of water was captured at 95 sites in the Attert basin between July 2015 and June 2018 using electric conductivity (EC) sensors. We used Onset HOBO Pendant waterproof temperature and light data logger (Model UA-002-64, Onset Computer Corp, Bourne, MA, USA) with modified light sensor to measure electric conductivity as proposed by

Chapin et al. (2013). The modified loggers were calibrated to determine EC from the recorded raw light intensity data (Lieder et al., 2017). Each sensor was covered in aluminium housing which protects the sensor against moving bedload and which also shades the sensor from direct sunlight to ensure unaffected water temperature measurements (Figure 3). The housing was mounted to boulders or concrete structures at the deepest point accessible within the channel, or attached with stainless steel ropes to trees or other suitable riparian structures. Sensors were fixed with zip ties inside the housing. This

setup enables fast access to the sensor during maintenance and data collection. Logging frequency was 10 minutes.

Stream sites for water temperature and EC measurements were selected to cover representatively the underlying geology and land use of the study region, as well as different stream and corresponding catchment sizes. For this method most sites were selected where streamflow was assumed to be continuous for most of the year. A total of 95 modified sensors were installed. 29 confluences were equipped with a setup of three sensors to capture the two tributaries and the mixing stream water and 8

stream sites were equipped with a single sensor to gain additional information on those stream points.

The collected dataset was filtered to gain information of the presence or absence of water at the sensor: recorded EC data of 0 to 25 µS cm$^{-1}$ indicated dry conditions. Based on the binary classification into presence or absence of water the range of EC between 1 and 25 µS cm$^{-1}$ was considered as dry conditions, although it might indicate a transition between dry and wet conditions, when sensors were still wet but not submerged in the river water anymore. Measured EC above 25 µSi cm$^{-1}$ was

considered to be 'flowing water' assuming that the water at the sensor is flowing. Data gaps due to failure of equipment or lost loggers are indicated by NA values.




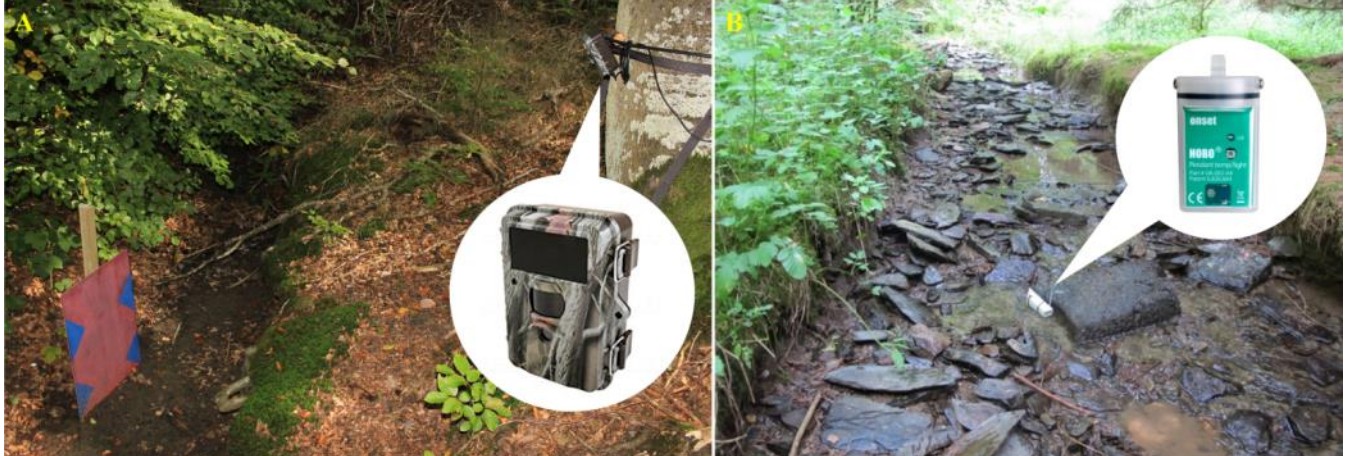

**Figure 3: Sensor setup for time lapse camera (left) and EC-sensor (right). The EC-sensor housing is mounted to a concrete block in the streambed.**

### 3.3 Streamflow observation with water level sensors

The described dataset includes 11 sites from the cluster sensor network of the CAOS project (Zehe et al. 2014) where stream water level was measured. Water level was monitored with METER/Decagon CTD pressure transducers in stilling wells at weirs at 5 minutes resolution. Binary information of presence or absence of flow was obtained from the water level data.

Outliers in the dataset were detected by comparing a moving window median filter to the original dataset. Outliers were removed from the data and are indicated as no-data values in the dataset. Values greater than zero were classified as conditions where streamflow is present and marked with the value 1 while values of zero or smaller represent no-flow conditions and are classified as 0.

Sensor clusters were installed successively during the initial phase of the CAOS project from 2012 to 2013 and run until

March 2018. Sites were selected based on the idea of hydrological response unites (HRU) and cover dominant HRUs across the Attert catchment (Zehe et al. 2014). Water level sensors were mounted at 7 sites in the slate area, 5 sites in the marl and 2 in the sandstone area. Due to large data gaps two sites in the slate area and one in the sandstone are not included in the final dataset.

Additional gauging data was provided from the Luxembourg Institute of Science and Technology (LIST). This dataset

includes seven sites with discharge data with 15 minute resolution obtained from conventional gauging systems at the outlet of the Attert catchment as well as the subcatchments of Colpach, Huewelerbach, Roudbaach, Schwebich, Weierbach and Wollefsbach. Sub-catchments were chosen based on their predominant geology. The Colpach represents a catchment dominated by slate geology of the Ardennes and includes the headwater catchment of Weierbach. The geology of the



Schwebich catchment is dominated by Keuper Marls and hosts the headwater catchment of the Wollefsbach. The Huewelerbach is a headwater catchment which geology mainly consists of the Luxembourg Sandstone formation. The Roudbaach catchment at Platen represents a catchment of mixed geological substrata (Pfister et al., 2017). Discharge values were transformed into no-flow (0) values when discharge was zero and to streamflow (1) if discharge values were above

5   zero.

An overview of all measurement approaches and corresponding information are provided in Table 1. The datasets with different temporal resolutions were homogenized to a 30 minutes interval. Binary values of flow/water presence (1) and absence (0) were temporal averaged for the 30 minutes time step. If the average value was ≥0.5 it was classified as flow (1) otherwise as no flow (0). The 15th and 45th minute of an hour were used for the time-stamp to represent the averaged values.

**Table 1: Overview of sensors included in the streamflow dataset.**

|  | ID in dataset | Sensor Type | Original Measurement Interval [minutes] | Datatype |
|---|---|---|---|---|
| Time-lapse cameras | C1 to C70 | Time-lapse cameras | 15 | presence/absence of streamflow [binary] |
| EC-sensors | EC1 to EC 95 | EC-sensors | 10 | presence/absence of water [binary] |
| CAOS-gauges | CG1 to CG11 | Conventional Gauges | 5 | stage/discharge |
| LIST-gauges | CG12 to CG18 | Conventional Gauges | 15 | discharge |

For the data presentation we classified the data into ephemeral, intermittent and perennial according to the classification scheme of Hedman and Osterkamp (1982). This classification is based on the proportional annual duration of streamflow

15   being present in a stream which in this study is defined as intermittency ratio ($I$). Sites are classified as ephemeral when $I <$ 0.2, intermittent when $0.2 \leq I < 0.8$ and perennial when $0.8 \leq I$.





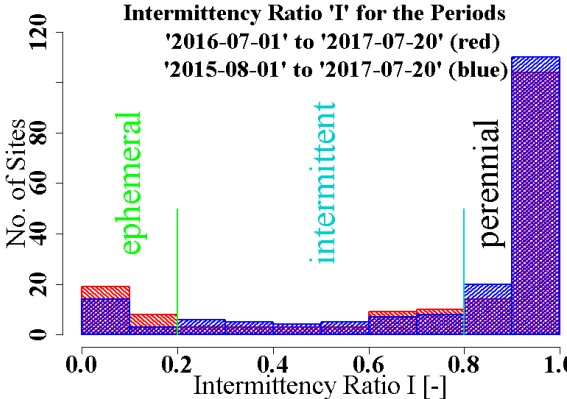

**Figure 4: Intermittency ratio of the periods 01.07.2016 – 20.07.2017 (red) and 01.08.2015 – 20.07.2017 (blue). According to the classification of Hedman & Osterkamp (1982) the degree of intermittency is indicated for ephemeral, intermittent and perennial streams. The distribution of classes shows a high number of perennial sites compared to ephemeral and intermittent.**

## 4 Data description

All applied methods successfully differentiated between presence and absence of streamflow or water. Sites were equipped for different periods depending on the method and purpose. Most of the time series (162 of 182) end in July 2017, with the longest time-series resulting from the conventional gauges starting from either January 2013 (7 gauges) or September 2013

10 (11 gauges); EC-data starts between August and November 2015 with a total number of 93 sites. 40 Time-lapse cameras were installed in August to November 2015 and additional 30 sites were equipped in June 2016.

For the overlapping period of August 2015 to July 2017 we measured the presence and absence of streamflow at 142 sites in the Attert basin (Figure 7). 129 of the sites show less than 20 % of no-data records during that time period (Figure 5). The best data coverage is given for the period July 2016 to July 2017 with 182 equipped sites. Data series with less than 20 % of

15 no-data records can be found in 155 of the time series for that time-period (Figure 5).



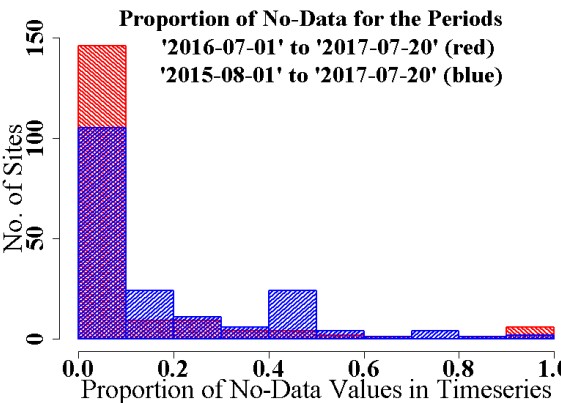

**Figure 5: The proportion of No-Data-values in each time series is shown for all sites. The two selected periods 01.07.2016 – 20.07.2017 and 01.08.2015 – 20.07.2017 represent the best data availability with a more complete data set for the shorter period with nearly 150 complete time series shown in red.**

Although time lapse cameras were primarily installed at ditches and channels with assumed ephemeral to intermittent streamflow, some of these sites showed continuous streamflow during the observation period. EC-measurements show permanent flow for most of the sites with few exceptions at sites which were chosen as reference site for intermittent streams observations with time-lapse cameras installed in nearby stream sections for validation. At most of the conventionally

10   gauged streams from the dataset provided by the LIST we find perennial streamflow whereas the sites equipped by the CAOS project show a higher degree of intermittency. Figure 6 shows the linkage between the proportions of bedrock geology found in the catchments of each monitoring site, the catchment area and the degree of intermittency within the period 01.08.2015 – 20.07.2017. Geology and catchment area are both known drivers of intermittency in the Attert catchment (Pfister et al. 2017).and therefore this additional information is included in the dataset. An overview of the

15   complete dataset featuring data from time-lapse camera (C), EC-sensors (EC) and conventional gauges (CG) is provided in Figure 7 and reveals fluctuation between wet and dry periods especially at sites monitored with time-lapse cameras.





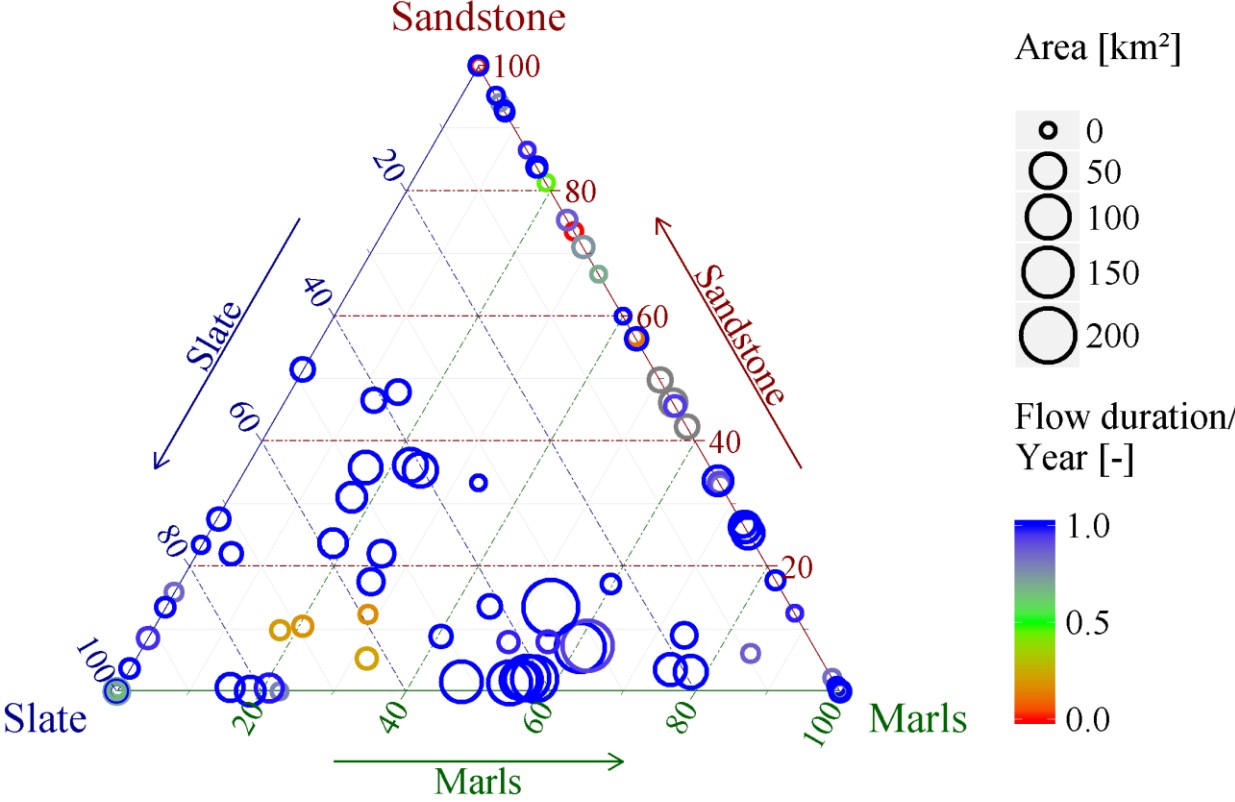

**Figure 6: Measurement sites in relation to the proportion of predominant bedrock geology within their catchments. Catchment area is indicated by the size of each circle while the flow duration per year is shown by the colour scale (grey colors missing data from downstream of the Attert gauge at Useldange). Larger catchments in the Attert basin typically have a higher proportion of slate and marl bedrock geology while smaler headwater catchments often host only one or two of the three predominant geologies.**



**Figure 7: Streamflow data from July 2015 - July 2017. Gauge IDs consist of the abbreviations "C" for time-lapse cameras, "EC" for EC-sensors and "CG" for conventional gauges on the right and the corresponding number of the gauge ID on the left. Data is ordered by gauge ID for each sensor type.**



## 5 Data quality and uncertainties

Although time lapse imagery generally provides very accurate information on the presence or absence of streamflow compared to the EC-sensor approach or even water level sensors, it has limitations when the quality of images is affected. Various sources can introduce noise, including direct impact of sunlight (Figure 8A), scattering light reflections on the

surface or gauging plate due to moving leaves and moisture in the lens system (Figure 8D). The view on the stream can be limited due to plant-growth during the growing season (Figure 8B), fog between lens and channel as well as spiders or insects in front of the lens Gaps in the time-series or additional data losses also happened due to vandalism (Figure 8C) or theft of equipment. Additionally, data gaps were caused by unintended resetting of the internal camera software after issues with the power supply. This problem increased towards the end of the data acquisition period as cable connections loosened

more frequently. Missing data in the time-series is marked as NA Value. Besides the issue of missing data classification of streamflow into binary categories of presence or absence can lead to different results when classified by different persons. This issue was tackled by training of persons involved in the visual image analysis with three example sequences of transitions from flow absence to presence and back with images taken under regular light conditions as well as scatter light conditions and images showing a litter filled streambed. Trained with these data samples and the according predefined

classification all involved image analysts got examples for all major conditions that were observed during the monitoring campaign which increased the homogeneity among the classification of different analysts. Finally, without motion picture one cannot distinguish between actual presence of flow and pooling water in a wet channel. The additional information of water flow would be beneficial compared to the information of water being presence when defining active channels as stated by Shaw (2016). In this study we assume all wet states of a channel to represent active flow conditions.





**Figure 8: Sources of uncertainties and errors in the time-lapse imagery data. A) reflections of sublight into the lens, B) growth of vegetation between camera and stream, C) damaged cable connection after attempted SD-card theft or D) by condensated water within the lens system.**

For the EC measurements to accurately describe streamflow presence or absence, sensor location is critical. Sensors not located at the lowest point of the stream can wrongly indicate a no-flow or no-water situation. With changes in streambed morphology following a high-flow event or anthropological disturbance, sensors might either end up in a dry stream segment with low flows bypassing the sensor, or in a puddle within an otherwise dry streambed. Filtering of the EC data to represent

10  flow/no-flow conditions was unproblematic due to the distinct and consistent differences in measured EC values, which can be used to accurately identify whether the sensory is completely dry, enclosed in wet but unsaturated sediment, or submerged in water.

Data quality of the conventional gauges was generally high but at gauges at CAOS sites, which were installed at smaller streams with a higher streamflow variability, issues with eroded weirs and altered channel cross section were reported. No



problems were reported with data obtained from the gauges maintained by the LIST with exceptions for data during low-flow conditions at freezing temperatures in the winter season 2017 which led to uninterpretable data during a short period leading to data gaps. The gauge at the Schwebich river was under maintenance after the winter 2017 and thus data from 2017 is missing.

**6 Data availability**

Data for presence and absence of streamflow for all presented sites are freely available from the online repository of the German Research Center for Geosciences (GFZ) (Kaplan et al., 2019) and has the DOI http://doi.org/10.5880/FIDGEO.2019.010. For each monitoring system (time-lapse cameras, EC-sensors, conventional gauges) one text file is available containing a time series starting from 01.01.2013 and ending on 17.07.2017 with a temporal

resolution of 30 minutes. We provide the location of all sites in shape-file format in the projected coordinate system Luxembourg 1930 Gauss (EPSG-WKID: 2169). The shapefile includes the site ID, share of bedrock geology for each watershed, catchment area and mean catchment slope. Included is a readme file that contains a detailed description of data file contents, including header information and contact information for additional details. Additional data for the Attert catchment is published in articles of this special issue and is available at the same repository.

**Author contributions**

Nils Kaplan and Ernestine Sohrt designed the experiments and carried them out. Nils Kaplan prepared the manuscript with contributions from all co-authors.

**Competing interests**

The authors declare that they have no conflict of interest.

**Acknowledgements**

This study was funded by the German Research Foundation (DFG) under the umbrella of the Research Unit FOR 1598 Catchments As Organized Systems (CAOS) – subproject G "Hydrological connectivity and its controls on hillslope and catchment scale stream flow generation". We like to thank the Luxembourg Institute of Science and Technology (LIST) for providing geospatial and discharge data. We appreciate the work of Cyrille Tailliez and Jean François Iffly who are

responsible for the installation, maintenance and data processing at the LIST and contributed with their work to our project. We thank Dominic Demand, Britta Kattenstroth, Tobias Vetter, André Böker, Jonas Freymüller and Eduardo Reppert for



their support during field work. We also thank Carolin Winter, Katrin Kühnhammer and Robin Schwemmle for their support with the image analysis.

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
