# Peer review of "Monitoring ephemeral, intermittent and perennial streamflow: A data set from 182 sites in the Attert catchment, Luxembourg"

_Earth System Science Data, 2019_

## Referee Comment (RC1) · Anonymous Referee #1 · 16 Apr 2019

General comments:

This paper describes the methods and logic behind an extensive data set that will be an important contribution to researchers studying intermittent and ephemeral streams. Understanding the conditions under which flow is initiated in streams and ditches, and are thus highly connected to their watersheds is an important field of research. Particularly, this study tests new methods against old methods and combines them to provide 182 monitored streams. This high-quality dataset will help many researchers test hydrological models of where and when streams flow, especially given different geology and topography.

[Figure]

Specific comments:

Page 7, Lines 18-21: When I first read this I was very excited by the Image J plugin and thought that it classified all photos as streamflow or non-streamflow. But, later when I read the "Data quality and uncertainties" section (page 15, line 10), I realized that there were people involved in this evaluation. I think it would be good to make this point more clear. Rephrase as something like, "Pre-processed images were loaded as virtual stacks in to ImageJ and then visually classified by trained interpreters..."

Technical corrections:

Page 7, line 12: In general, one should not start sentence with a number ("3" in this case), rather, spell out the word "Three" or restructure the sentence so that it does not begin with a number. – check this throughout the manuscript (i.e. page 11 there are a few).

Page 7, line 15: Super cool to have an Image J plugin! But... see specific comments above

Page 12, line 14: There is a period instead of a space after the (Pfister et al. 2017).

Page 15, lines 4-7: Typically, figures are referred to in their order of appearance. It is strange to jump from 8A to 8D. Please order them in the order you discuss them.

---

## Referee Comment (RC2) · Anonymous Referee #2 · 23 Apr 2019

General comments

While this is registered as a research article, it is in fact a dataset of stream flow presence/absence. The dataset comprises three text files with data series of stream flow presence/absence, measured with three different methods (monitored by time-lapse cameras, by EC-sensors and by conventional gauges). There is also a shape-file with the measurements sites, also containing attribute data for each catchment. The data files and the methodology are described in a PDF file with metadata. My first thought is that I applaud the idea of publishing such a data set. The science community know little about small-scale streams/ephemeral/intermittent streams and field measurements

are often time consuming and costly. While we often use models to advance our understanding of the hydrology field datasets are necessary to validate models. This dataset therefore fills an important function and I believe the dataset should be published.

Specific comments

Timeseries

I don't have any comments on the three text files with time-series data. I found the files well organized and easy to understand, given the information in the PDF metadata file.

Geodata

When it comes to the geodata I have some questions regarding the description of the file. The file contains catchment area (c_area). I'm missing the unit. m2, ha, sqr feet? How was it calculated? Using a DEM? Is it possible to also share the catchment boundaries? I always find that that helps with the overview and understanding of a dataset, in a much more intuitive way than attributes in a point-file. slope_avg; I'd like a more thorough description in the pdf of the average slope, average of what? How was slope calculated? Based on a DEM, what resolution? In radians or degrees?

Other than the clarification of c_area and slope_avg I don't have comments on the dataset and I believe the dataset should be published.

———————————————

---

## Referee Comment (RC3) · Anonymous Referee #3 · 9 May 2019

General comments:

While there are plenty of datasets of time series with streamflow from permanent streams there is a need to address ephemeral and intermittent streams. This paper address this issue by presenting a novel dataset of presence and absence of stream flow within a 247 km$^2$ catchment in Luxembourg. The quality and time resolution of the dataset is high and the data can be used to evaluate hydrologic models.

Specific comments:

The introduction can be shortened and more focused on the methods used for this dataset.

[Figure]

Bedrock, land use, climate and topography are well described. However a sentence Explaining the surface soils would be useful to list under site description.

The shapefile that contains the spatial data there are different attributes summarized within catchments. In order for someone else to work with this data it's important to have access to these catchments. I suggest that a description of how these catchments were extracted is included. Perhaps even upload the catchments along with the streamflow data.

I'm not clear on how exact the locations of the points are. Any estimate of GPS accuracy would help future users when delineating catchments or conducting hydrological modeling.

Technical corrections:

Minor comments on figure 1: North arrows are only required if the map is not north-oriented. It can be removed to clean up the look of figure 1.

Figure 4 and 5: in the figure legend dates are written 'year-month-day" but in the text it's written "day-month-year". Is this intentional?

Figure 6 description, bottom line: check spelling of "smaler". Should probably be "Smaller". Figure 6: The labels for proportion of sandstone is written horizontally while the labels for marls and slate are tilted. I suggest that you display all labels horizontally. Otherwise a neat figure.

---

## Author Response (AR1)

Dear Referee,

Thank you for your helpful comments and questions to our manuscript. We appreciate your recommendation for the publication of the dataset. Please find your questions and comments marked as e.g. << R1.C1: question/comment>> followed by our answer marked as e.g. R1.A1: below.

Best regards,

Kaplan et al.

<< R1.C1:
"Specific comments:
Page 7, Lines 18-21: When I first read this I was very excited by the Image J plugin and thought that it classified all photos as streamflow or non-streamflow. But, later when I read the "Data quality and uncertainties" section (page 15, line 10), I realized that there were people involved in this evaluation. I think it would be good to make this point more clear. Rephrase as something like, "Pre-processed images were loaded as virtual stacks in to ImageJ and then visually classified by trained interpreters…"
>>

R1.A1: Adding the information about trained interpreters here will definitely clarify the procedure used to generate the data and we will add this to the revised manuscript. We will include a sentence as proposed.
In addition to the pre-processor plugin used in this study we developed a second plugin with the aim to identify the water line on the plate and calculating the water level by making use of automated thresholding functions of ImageJ followed by a Hough transformation to identify the water line. This method was developed for one site witch was located in grassland without interference of light scatter or shading on the plate by vegetation. This approach showed promising results, without 100% accuracy though. Nonetheless, under conditions of changing shading and light scatter on the gauging plate at forest sites, no reliable waterline detection could be achieved, even with some further testing of different auto-thresholds. This second plugin development is not part of the paper because the data was not generated with the use of the plugin. We think that information about a plugin that was not used to generate the data would confuse the reader more than it would inform.

In addition we identified some major drawbacks of the gauging plate design in our study, 1) The wooden plates are good for cheap testing phases, but will not endure for more than two years; 2) The handmade paint on the plate looks like an even layer but when it comes to image analysis slight differences in the paint thickness become visible and interfere with the line detection algorithm; 3) The design of the blue detection patterns on the plate need to be improved to be fully able to serve as fiducial markers in the image analysis. Therefore, we used the more reliable method with trained interpreters to produce high quality data.

R1.C1-changes: We included the information "by trained interpreters" (page 7, line 23) as suggested by the referee.

<< R1.C2:

Technical corrections:
Page 7, line 12: In general, one should not start sentence with a number ("3" in this case), rather, spell out the word "Three" or restructure the sentence so that it does not begin with a number. – check this throughout the manuscript (i.e. page 11 there are a few).
Page 7, line 15: Super cool to have an Image J plugin! But… see specific comments above
Page 12, line 14: There is a period instead of a space after the (Pfister et al. 2017).
Page 15, lines 4-7: Typically, figures are referred to in their order of appearance. It is strange to jump from 8A to 8D. Please order them in the order you discuss them.
\>\>

R1.A2: Thank you very much for your helpful technical corrections, we will make all the suggested modifications when revising the manuscript

R1.C2-changes: We included the technical corrections that were state above. Figure 8 was reordered to the order it is discussed in the text. The letters were changed to small letters in brackets according to ESSD standards.

*R2:*
Thank you for your helpful comments and questions to our manuscript. We appreciate your recommendation for the publication of the dataset. Please find your questions and comments marked as e.g. << R2.C1: question/comment>> followed by our answer marked as e.g. R2.A1: below.

Best regards,

Kaplan et al.

<< R2.C1:
Geodata
When it comes to the geodata I have some questions regarding the description of the file.
>>

R2.A1: Thank you for your justified questions regarding the information of the geodata. We will clarify the data description and update a revised data description. My suggestion is to update the "Methods" section in the data-description with a new paragraph "4.4 Geodata" with a detailed description of the geodata processing. Please find your questions answered below.

R2.C1-changes: We added the paragraph "4.4 Geodata" in the "4. Methods" section of the data-description including a detailed description of the geodata processing. This comprises changes based upon several of the following comments of this referee. The full section is included as follows:

"Geodata comprises of information on proportional shares of geological units in the catchment, the average slope in the catchment and the catchment area upstream of each site. Geological information is derived from a geological map (1:25.000) provided by the Administration des ponts et chaussées Service géologique de l'Etat, Luxembourg (2012). The the original map was created from 1947-1949. GIS analyses were performed using QGIS and SAGA on a 15 m resolution digital elevation model (DEM), which is based on a combined 5m resolution LIDAR scan of Luxembourg (Modèle Numérique de Terrain de Luxembourg, Le Gouvernement du Grand-Duché de Luxembourg, Administration du cadastre et de la topographie, 5m LIDAR, https://data.public.lu/en/datasets/bd-l-mnt5/) and 10m resolution LIDAR scan of Belgium (Relief de la Wallonie - Modèle Numérique de Surface, Service public de Wallonie, Département de la Géomatique. 10m LIDAR, http://geoportail.wallonie.be/catalogue/6029e738-f828-438b-b10a-85e67f77af92.html). The generated 15m DEM has been pre-processed by burning in the digitalized stream network ( min. border cell method, epsilon = 3) and filling sinks (Wang Lui algorithm, minimum slope = 0.1°). The catchment area was calculated by using the pre-processed DEM with 15m resolution and the catchment area recursive tool from the SAGA toolbox using the D-8 method. The same DEM was used to calculate the average slope of each catchment. The "slope, aspect, curvature" tool from the SAGA toolbox was used to calculate the slope [radians] with the 9 parameter 2nd order polynom method (Zevenbergen & Thorne 1987) which uses a 3x3 pixel window of the DEM to calculate the slope. Catchment boundaries for each site are included as shape files. These shapefiles were calculated with the Watershed tool from the ArcGIS Hydrology toolbox using a flow direction raster as input which was derived from the Flow Direction tool (ArcGIS Hydrology toolbox) from the DEM described

above. Raster output was transformed to shape files without simplification of the geometry (sub-folder: boundaries)."

<< R2.C2:
The file contains catchment area (c_area). I'm missing the unit. m2, ha, sqr feet?
>>
R2.A2: The catchment area's unit is m$^2$, we will add it to the data-description.

R2.C2-changes: We added the unit m² in the data-description in the section 6. File Description, Subsection 6.1 Folder: Geodata.

<< R2.C3:
How was it calculated? Using a DEM?
>>

R2.A3: GIS analyses were performed using QGIS and SAGA on a 15 m resolution digital elevation model (DEM), which is based on a combined 5m resolution LIDAR scan of Luxembourg (Modèle Numérique de Terrain de Luxembourg, Le Gouvernement du Grand-Duché de Luxembourg, Administration du cadastre et de la topographie, 5m LIDAR, https://data.public.lu/en/datasets/bd-l-mnt5/) and 10m resolution LIDAR scan of Belgium (Relief de la Wallonie - Modèle Numérique de Surface, Service public de Wallonie, Département de la Géomatique. 10m LIDAR, http://geoportail.wallonie.be/catalogue/6029e738-f828-438b-b10a-85e67f77af92.html). The generated 15m DEM has been pre-processed by burning in the digitalized stream network ( min. border cell method, epsilon = 3) and filling sinks (Wang Lui algorithm, minimum slope = 0.1°). The catchment area was calculated by using the pre-processed DEM with 15m resolution and the catchment area recursive tool from the SAGA toolbox using the D-8 method.

R2.C3-changes: We included the calculations of catchment area using GIS in the description file, section 4. "Methods", sub-section "4.4 Geodata". "The catchment area was calculated by using the pre-processed DEM with 15m resolution and the catchment area recursive tool from the SAGA toolbox using the D-8 method."

<< R2.C4:
Is it possible to also share the catchment boundaries?
>>

R2.A4: We never calculated catchment boundaries of complete (nested) sub-catchments. We calculated the catchment boundaries using the "Watershed" tool from the ArcGIS toolbox. This results in a raster layer showing the areas which belong to each part of a sub-catchment between the pour point and the corresponding upstream pour point(s). Without the information of stream network and the longitudinal topology between the sub-catchments the data does not enhance visual exploration of the data. Although to our knowledge there is no standard GIS function available which derives individual catchment boundaries for multi points, we will try to add a dataset with shape-files for each site containing the catchment boundaries.

R2.C4-changes: We included the catchment boundaries as single shape files named according to their corresponding Gauge ID. The information about the watersheds was included in the description file, section 4. "Methods", sub-section "4.4 Geodata". The information that this data is also available is included in the data availability section of the manuscript.

<< R2.C5:
slope_avg; I'd like a more thorough description in the pdf of the average slope, average of what?
>>

R2.A5:
This is indeed not self-explanatory. We will include a more detailed description in the new paragraph "4.4 Geodata". Slope average is the average slope of the sub-catchment for the specific site.

R2.C5-changes: We changed the wording in section 6. File Description, Subsection 6.1 Folder: Geodata from "Average slope in watershed upstream of monitoring site" to "Average slope [rad] of the watershed upstream of monitoring site".

<< R2.C6:
How was slope calculated?
>>

R2.A6: The "slope, aspect, curvature" tool from the SAGA toolbox was used to calculate the slope with the 9 parameter $2^{nd}$ order polynom method (Zevenbergen & Thorne 1987) which uses a 3x3 pixel window of the DEM to calculate the slope. Degree and radians were both calculated. In this geodata the radians have been included. We will add this information to the description file.

R2.C6-changes: We included the calculations of the average slope using GIS in the description file, section 4. "Methods", sub-section "4.4 Geodata". "The same DEM was used to calculate the average slope of each catchment. The "slope, aspect, curvature" tool from the SAGA toolbox was used to calculate the slope [radians] with the 9 parameter $2^{nd}$ order polynom method (Zevenbergen & Thorne 1987) which uses a 3x3 pixel window of the DEM to calculate the slope."

<< R2.C7:
Based on a DEM, what resolution?
>>

R2.A7: The same DEM which has been used to calculate the catchment area was used to calculate the average catchment slope. We will clarify this in the revised manuscript.

R2.C7-changes: The clarification is not in the manuscript but in the data description, section 4. "Methods", sub-section "4.4 Geodata". The total process of calculating the slope in GIS is included (see also R2.C6).

<< R2.C8:
In radians or degrees?
>>

R2.A8: Radians are included in this dataset. Including also degrees would be convenient, but can easily be calculated from radians.

R2.C8-changes: We changed the wording in the data description, section 6. File Description, Subsection 6.1 Folder: Geodata from "Average slope in watershed upstream of monitoring site" to "Average slope [rad] of the watershed upstream of monitoring site". The use of radians is also included in the description of the slope calculation in the new section "4.4 Geodata".

**R3**

**Interactive comment on* "Monitoring ephemeral, intermittent and perennial streamflow: A data set from 182 sites in the Attert catchment, Luxembourg"**

Dear Referee,

Thank you for your helpful comments and questions to our manuscript. We appreciate your recommendation for the publication of the dataset. Please find your questions and comments marked as e.g. << R3.C1: question/comment>> followed by our answer marked as e.g. R3.A1: below.

Best regards,

Kaplan et al.

Specific comments:
<< R3.C1:
The introduction can be shortened and more focused on the methods used for this
dataset.
>>

R3.A1: Thank you for your comment on the length of the introduction. We will check for potential parts of the text that can be shortened. The introduction is structured into a short introduction to the classification of ephemeral, intermittent and perennial streams in either statistical or physical based classification methods, followed by a short overview of potential methods that are used to observe streamflow. We put a little more emphasis on the statistical classification methods because we apply them for the data descriptions in the data section. Still, we see a potential to shorten the section of statistical based classification methods. For us it is important to show that methods for streamflow monitoring developed over time and recently a variety of methods evolved from many different studies which all have benefits and drawbacks in their specific way. With our choice of methods and the existing dataset we show that this combined setup of sensors is a functioning, yet not perfect setup for the continuous streamflow observation but the introduction should also offer space for those methods that are also available and might be considered when setting up a similar experiment in the future. Therefore, we prefer to keep this section but will shorten it if the editor agrees with the reviewer on this point.

R3.C1-changes: Page 1, line 26: We removed the sentence giving a detailed statistical classification from Matthews (1988).

<< R3.C2:
Bedrock, land use, climate and topography are well described. However a sentence
explaining the surface soils would be useful to list under site description.
>>

R3.A2: We agree and will add information about the surface soils in the site description.

R3.C2-changes: We added two sentences explaining briefly the soil patterns in the Attert basin on page 4 line 27.

"Surface soil patterns in the Attert basin are strongly related to lithology, land cover and land use (Cammeraat et al., 2018). Stagnosols to Planosols are prevailing soils on the marls, Leptosols, Arenosols and Podzols are predominantly found in the Luxembourg and Buntsandstein sandstone, while soils on Devonian slate comprise mainly of Cambisols (Martinez-Carreras et al., 2012)."

<< R3.C3:
The shapefile that contains the spatial data there are different attributes summarized within catchments. In order for someone else to work with this data it's important to have access to these catchments. I suggest that a description of how these catchments were extracted is included. Perhaps even upload the catchments along with the streamflow data.
>>

R3.A3: We refer here to our reply to the Anonymous Reviewer #2 (R2.A4). We used a DEM of 15m resolution in this study to calculate catchment size with the catchment area recursive algorithm from the SAGA GIS toolbox. We will update the Data-Descripition.pdf file with a more detailed description of the extraction of GIS  and will supply the shape files of the catchments.

R3.C3-changes: We added a new dataset of shapefiles including the catchment boundaries to the "Geodata" folder. This part of the dataset is also described in the section 4.4 Geodata in the data description.

<< R3.C4:
I'm not clear on how exact the locations of the points are. Any estimate of GPS accuracy would help future users when delineating catchments or conducting hydrological modeling.
>>

R3.A4: We used hand held GPS devices to identify the locations. Spatial accuracy of the device was sometimes at 8-10m, especially in the deep inclined valleys of the Ardennes and Luxembourg Sandstone, but generally between 3 and 4 m. We used a digital topographic map (1:20.000) to correct locations in the shape file if needed. This information will be added to the specific sections in the paper.

R3.C4-changes: We added the information on page 7, line 14 in the manuscript.

"The exact geographical position of each location was measured with a hand held GPS device (Garmin GPSmap 62s) which has a spatial accuracy between 3-4 m. The accuracy drops especially in the deep inclined valleys of the Ardennes and Luxembourg Sandstone to 8-10 m. We used a topographic map at the scale of 1:20.000 (Le Gouvernment du Grand-Duché de Luxembourg, 2009) to correct locations with low accuracy."

Technical corrections:
<< R3.C5:
Minor comments on figure 1: North arrows are only required if the map is not northoriented. It can be

removed to clean up the look of figure 1.
\>\>

R3.A5: We agree that clean looking figures improve every publication. However, the detail of including a North arrow in a map/figure is discussed in the scientific community but will never be debated in a cartography class. Prof. Dr. David Schultz (http://eloquentscience.com/2010/02/when-to-use-north-arrows-on-maps/) presents some examples, where an added North arrow not only helps to identify the orientation of the map but could also help to identify erroneous data. He states: "The addition of a north arrow can never harm a figure, only help with clarity". Some journals advise to include a North arrow in figures (e.g.: https://onlinelibrary.wiley.com/page/journal/20544049/homepage/forauthors.html), whereas ESSD and HESS leaves the decision open to the authors and editors as clear guidelines are missing. In this case we would prefer to follow the recommendation of Prof. Schultz and keep the North arrow.

R3.C5-changes: we keep the figure as is according to the explanation given above.

\<\< R3.C6:
Figure 4 and 5: in the figure legend dates are written 'year-month-day" but in the text
it's written "day-month-year". Is this intentional?
\>\>

R3.A6: We will change the date in the figure to the ESSD standard dd.mm.yyyy.

R3.C6-changes: We changed the date in the figures 4 and 5 according to the ESSD standard (page 11, line 1; page 12; line 1).

\<\< R3.C7:
Figure 6 description, bottom line: check spelling of "smaler". Should probably be
"Smaller". Figure 6: The labels for proportion of sandstone is written horizontally while
the labels for marls and slate are tilted. I suggest that you display all labels horizontally.
Otherwise a neat figure.
\>\>

R3.A7: We will change the spelling to the correct "smaller". The labels in Figure 6 are intentionally tilted and non tilted according to the dashed lines which delineate the percentages in the plot. We used the ggtern library in R to generate the plot.

R3.C7-changes: We changed the spelling to "smaller" in the captions of Figure 6. The design of the figure has not been changed as discussed above.

[revised manuscript text omitted]

Zehe E., Ehret U., Pfister L., Blume T., Schröder B., Westhoff M., Jackisch C., Schymanski S.J., Weiler M., Schulz K., Allroggen N., Tronicke J., van Schaik L., Dietrich P., Scherer U., Eccard J., Wulfmeyer V. and Kleidon A.: HESS Opinions: From response units to functional units: a thermodynamic reinterpretation of the HRU concept to link spatial organization and functioning of intermediate scale catchments, Hydrol. Earth Syst. Sci., 18, 4635-4655, doi: 10.5194/hess-18-4635-2014, 2014.

**Time series of streamflow occurrence from 182 sites in ephemeral, intermittent and perennial streams in the Attert catchment, Luxembourg.**
**(http://doi.org/10.5880/fidgeo.2019.010)**

**1. Licence and Citation**

These data are freely available under a Creative Commons Attribution 4.0 International (CC BY 4.0) Licence. When using the data please cite:

Kaplan, Nils Hinrich; Sohrt, Ernestine; Blume, Theresa; Weiler, Markus (2019): Time series of streamflow occurrence from 182 sites in ephemeral, intermittent and perennial streams in the Attert catchment, Luxembourg. V. 1.0. GFZ Data Services. http://doi.org/10.5880/FIDGEO.2019.010

**The data are supplementary material to:**

Kaplan, Nils Hinrich; Sohrt, Ernestine; Blume, Theresa; Weiler, Markus (2019): Monitoring ephemeral, intermittent and perennial streamflow: A data set from 182 sites in the Attert catchment, Luxembourg. Earth System Science Data Discussion Paper, https://doi.org/

**2. Authors and affiliations:**

Nils Hinrich Kaplan[1], Ernestine Sohrt[2], Theresa Blume[2], Markus Weiler[1]

1. Hydrology, Faculty of Environment and Natural Resources, University of Freiburg, Freiburg, Germany
2. Section Hydrology, GFZ German Research Centre for Geosciences, Potsdam, Germany

Correspondence to: N. Kaplan (nils.kaplan@hydrology.uni-freiburg.de)

**3. Data Description**

We used different sensing techniques including time-lapse imagery, electric conductivity and stage measurements to generate a combined dataset of presence and absence of streamflow within a large number of nested sub-catchments in the Attert Catchment, Luxembourg. The first sites of observation were established in 2013 and successively extended to a total number of 182 in 2016 as part of the project "Catchments As Organized Systems" (CAOS, Zehe et al., 2014). Setup for time-lapse imagery measurements was inspired by Gilmore et al. (2013) while the setup for EC-sensor was proposed by Chapin et al. (2014). Temporal resolution ranged from 5 to 15 minutes intervals. Each single dataset was carefully processed and quality controlled before the time interval was homogenized to 30 minutes. The dataset provides valuable information of the dynamics of a meso-scale stream network in space and time.

The Attert basin is located in the border region of Luxembourg and Belgium and covers an area of 247 km². The elevation of the catchment ranges from 245 m a.s.l. in Useldange to 549 m a.s.l. in the Ardennes. Climate conditions across the catchment are rather similar in terms of temperature and precipitation. Hydrological regimes are mainly driven by seasonal fluctuations in evapotranspiration causing flow to cease in intermittent reaches during dry periods. The catchment covers three predominant geologies: Slate, Marls and Sandstone. The dataset features data from catchments covering all geological characteristics from single geology to mixed geology. It can be used to test and evaluate hydrologic models, but also for the assessment of the intermittent stream ecosystem in the Attert basin.

**4. Methods**

**4.1 Time-lapse Imagery**

Dörr Snapshot Mini 5.0 consumer wildlife cameras were used for time-lapse imagery. Time lapse monitoring was realized with the internal software with a temporal resolution of 15 minutes. Cameras were mounted at trees or structures close to the channel. For improved image analysis a gauging plate was installed in the channel. This method was closely related to a time-lapse camera gauging system published by Gilmore et al. (2013).

**4.2 EC-sensors**

Onset HOBO Pendant waterproof temperature and light data logger (Model UA-002-64, Onset Computer Corp, Bourne, MA, USA) with modified light sensor to measure electric conductivity were used to monitor electric conductivity (EC) as proposed by Chapin et al. (2014). EC values were classified into no-flow situations for EC-values below 25microSi/cm and flow situation for EC-values above 25microSi/cm.

**4.3 Conventional Gauges**

Conventional Gauges are divided into two sub-datasets. Data from ID values CG1 to CG11 were derived from water level data measured by METER/Decagon CTD pressure transducers in stilling wells. Data from ID values CG 12 to CG 18 were derived from discharge values measured by the Luxembourg Institute of Science and Technology (LIST).

**4.4 Geodata**

Geodata comprises of information on proportional shares of geological units in the catchment, the average slope in the catchment and the catchment area upstream of each site. Geological information is derived from a geological map (1:25.000) provided by the Administration des ponts et chaussées Service géologique de l'Etat, Luxembourg (2012). The the original map was created from 1947-1949. GIS analyses were performed using QGIS and SAGA on a 15 m resolution digital elevation model (DEM), which is based on a combined 5m resolution LIDAR scan of Luxembourg (Modèle Numérique de Terrain de Luxembourg, Le Gouvernement du Grand-Duché de Luxembourg, Administration du cadastre et de la topographie, 5m LIDAR, https://data.public.lu/en/datasets/bd-l-mnt5/) and 10m resolution LIDAR scan of Belgium (Relief de la Wallonie - Modèle Numérique de Surface, Service public de Wallonie, Département de la Géomatique. 10m LIDAR, http://geoportail.wallonie.be/catalogue/6029e738-f828-438b-b10a-

85e67f77af92.html). The generated 15m DEM has been pre-processed by burning in the digitalized stream network ( min. border cell method, epsilon = 3) and filling sinks (Wang Lui algorithm, minimum slope = 0.1°). The catchment area was calculated by using the pre-processed DEM with 15m resolution and the catchment area recursive tool from the SAGA toolbox using the D-8 method. The same DEM was used to calculate the average slope of each catchment. The "slope, aspect, curvature" tool from the SAGA toolbox was used to calculate the slope [radians] with the 9 parameter 2nd order polynom method (Zevenbergen & Thorne 1987) which uses a 3x3 pixel window of the DEM to calculate the slope. Catchment boundaries for each site are included as shape files. These shapefiles were calculated with the Watershed tool from the ArcGIS Hydrology toolbox using a flow direction raster as input which was derived from the Flow Direction tool (ArcGIS Hydrology toolbox) from the DEM described above. Raster output was transformed to shape files without simplification of the geometry (subfolder: boundaries).

**5. Technical details of measurement setup**

**5.1 Time-lapse imagery equipment used**
- Dörr Snapshot Mini 5.0
- Mounting gear
- Cable lock
- Wire Lock for SD-security
- SD-card 16 GB<
- Battery pack "Hückmann Fiamm Lead-accumulator FG 10451"

**5.2 EC-measurements**
- Onset HOBO Pendant waterproof temperature and light data logger (Model UA-002-64, Onset Computer Corp, Bourne, MA, USA) with modified light sensor to measure electric conductivity
- Aluminum housing for shading
- Bolts

**5.3 Water level measurements**
- METER/Decagon CTD pressure transducers
- stilling wells

**6. File Description**
The dataset is divided into two folders: Geodata (geospatial information) and Timeseries.

**6.1 Folder: Geodata**
Geodata information includes one file (Binary_Streamflow_Data_Locations.shp) in shape format in the projected coordinate system Luxembourg 1930 Gauss (EPSG-WKID: 2169); The sub-folder "boundaries" includes shapefiles of catchment boundaries named corresponding to the Gauge

ID field of the file "Binary_Streamflow_Data_Locations.shp" in the Geodata folder. All geodata is provided in the projected coordinate system Luxembourg 1930 Gauss (EPSG-WKID: 2169).

The  file Binary_Streamflow_Data_Locations.shp includes following fields:

| FID | Field ID |
|---|---|
| Shape | Point features |
| X_LUREF | Latitude of the monitoring site in EPSG-WKID: 2169 reference system |
| Y_LUREF | Longitude of the monitoring site in EPSG-WKID: 2169 reference system |
| Gauge ID | ID of monitoring site identical with ID of monitoring sites in timeseries |
| Slate | Proportion of slate geology in watershed upstream of monitoring site; values 0 to 1; missing values -99 |
| Calcaire | Proportion of calcaire geology in watershed upstream of monitoring site; values 0 to 1; missing values -99 |
| Sandstone | Proportion of sandstone geology in watershed upstream of monitoring site; values 0 to 1; missing values -99 |
| Limestone | Proportion of slate geology in watershed upstream of monitoring site; values 0 to 1; missing values -99 |
| Alluvium | Proportion of alluvium geology in watershed upstream of monitoring site; values 0 to 1; missing values -99 |
| Marls | Proportion of marls geology in watershed upstream of monitoring site; values 0 to 1; missing values -99 |
| slope_avg | Average slope [rad]  of the watershed upstream of monitoring site; values > 0; missing values -99 |
| c_area | Catchment area upstream of monitoring site [m²] ; values > 0; missing values -99 |

**6.2 Folder: Timeseries**

Timeseries include binary of streamflow/water presence (1) and absence (0), 30 minutes temporal resolution, Timezone is UTC+1.

**FlowNoFlow_C.txt**: Streamflow presence/absence monitored by time-lapse cameras, starting date of data: 01.01.2013, date of first observations: 01.08.2015; file contains 70 columns: 1st column includes date and time, following columns contain one site per column, column name contains gauge ID.

**FlowNoFlow_EC.txt**: Water presence/absence monitored by EC-sensors, starting date of data: 01.01.2013, date of first observations: 01.07.2015; file contains 96 columns: 1st column includes date and time, following columns contain one site per column, column name contains gauge ID.

**FlowNoFlow_CG.txt:** Streamflow presence/absence monitored by conventional gauges, starting date of data: 01.01.2013, date of first observations: 01.01.2013; file contains 19 columns: 1st column includes date and time, following columns contain one site per column, column name contains gauge ID.